# Exploring the complex free-energy landscape of the simplest glass by rheology

Yuliang Jin[1] & Hajime Yoshino[1,2]

For amorphous solids, it has been intensely debated whether the traditional view on solids, in terms of the ground state and harmonic low energy excitations on top of it, such as phonons, is still valid. Recent theoretical developments of amorphous solids revealed the possibility of unexpectedly complex free-energy landscapes where the simple harmonic picture breaks down. Here we demonstrate that standard rheological techniques can be used as powerful tools to examine nontrivial consequences of such complex free-energy landscapes. By extensive numerical simulations on a hard sphere glass under quasistatic shear at finite temperatures, we show that above the so-called Gardner transition density, the elasticity breaks down, the stress relaxation exhibits slow, and ageing dynamics and the apparent shear modulus becomes protocol-dependent. Being designed to be reproducible in laboratories, our approach may trigger explorations of the complex free-energy landscapes of a large variety of amorphous materials.

[1] Cybermedia Center, Osaka University, Toyonaka, Osaka 560-0043, Japan. [2] Graduate School of Science, Osaka University, Toyonaka, Osaka 560-0043, Japan. Correspondence and requests for materials should be addressed to Y.J. (email: jinyuliang@cp.cmc.osaka-u.ac.jp).

Amorphous and crystalline solids have very different behaviours under external perturbations, especially rheological properties under shear deformations[1–18]. It is well known that by increasing the shear strain, a crystal displays a linear elastic response, followed by plastic deformation and yielding. However, experiments and numerical simulations show that this picture breaks down for amorphous solids, such as glasses[2,5–8,13,16,18,19], granular matter[3,11,15] and foams[4], where the elastic behaviour is mixed with plastic events. Such plastic events cause sudden drops in stress–strain curves, and are sometimes referred to as crackling noise[20], due to their similarities to avalanches in earthquakes. An apparent shear modulus or rigidity $\mu$, which is the ratio between the stress and strain, can be nevertheless defined and measured. Experiments on glassy emulsion systems[21,22] show that $\mu$ scales linearly $\mu \sim P$ with the pressure $P$ both below and above the jamming density, while harmonic treatments predict $\mu \sim P^{1.5}$ (below)[23] and $\mu \sim P^{0.5}$ (above)[24], respectively. These contradictions reveal that amorphous solids can be strikingly softer than purely harmonic solids like crystals, even at sufficiently low temperatures where the harmonic expansion was conventionally expected to be valid.

On the theoretical side, the mean-field theory based on the exact solution in the large dimensional limit of the hard sphere glass has brought a more accurate and comprehensive picture beyond the harmonic description[10,14,25–30]. The main outcome is the prediction of a Gardner transition (see Fig. 1a) that divides the classical amorphous phase into two: in the stable phase (or normal phase), the state is confined in one of the simple smooth basins on the free-energy landscape; once the system is compressed above the Gardner transition density $\varphi_G$ (or is cooled down below the Gardner transition temperature $T_G$), the simple glass basin splits into a fractal hierarchy of subbasins and the glass state becomes marginally stable. Although similar ideas of complex energy landscapes have been conceived phenomenologically in earlier works (see ref. 31 and references therein), the mean-field theory gives a firmer first principle ground for such a picture, with falsifiable predictions. In particular, the theory predicts that the elastic anomalies and nontrivial rheology should only appear in the marginally stable phase (or Gardner phase)[10,14] that lies deep inside the glassy phase. However, the mean-field theory is exact only in the large dimensional limit, and its relevance in real systems is far from obvious. Here we test the theoretical proposal of the nontrivial rheology in physically relevant dimensions $d = 2$ and 3 and compare quantitatively the theoretical predictions with our numerical data.

We design laboratory-reproducible rheological protocols to examine the signatures of the intriguing complex free-energy landscape. Our protocols are applied on densely packed hard spheres, a simple and representative glass-forming model. Our result shows the anticipated anomalous rheology emerging at the Gardner transition that turns out to be strikingly similar to the dynamical responses of spin glasses to an external magnetic field[32,33]. The evidence of a complex free-energy landscape in the Gardner phase is consistent with a previous numerical study[34] where particles' vibrational dynamics is analysed. That approach has been used in a recent experiment of an agitated granular system[35]. However, generalizing the method to other systems, such as molecular glasses, may not be easy due to the difficulty of tracking trajectories of individual particles. The approach proposed in this study overcomes this problem, since it requires no microscopic information, but only the standard macroscopic rheological measurements (the shear stress and strain) that are well accessible in many experimental systems, including molecular and metallic glasses, polymers and colloids. In the present paper, however, we do not attempt to judge whether the Gardner transition survives in finite dimensional systems as a sharp phase transition or becomes a crossover (in the thermodynamic limit), but rather we aim to explore the possibilities to observe its nontrivial signatures in experimentally feasible length/timescales.

## Results

**Preparation of stable glasses**. To avoid crystallization, we work on a polydisperse mixture of hard spheres whose diameters are distributed according to a probability distribution $P(D) \sim D^{-3}$, for $D_{min} \leq D < D_{min}/0.45$ (refs 34,36) (see Supplementary Note 1). A glass is typically obtained by a slow compression (or cooling) annealing from a dilute state, where it falls out of equilibrium at the compression (or cooling) rate-dependent glass transition density $\varphi_g$ (or glass transition temperature $T_g$). Since we choose hard spheres as our working system, the density is the control parameter.

We design a numerical protocol to mimic a simple shear experiment of deeply annealed glasses (see Fig. 1). Our protocol includes three steps. We first use the swap algorithm[35,36] to prepare a well-equilibrated, supercooled-liquid configuration at various densities $\varphi_g$ (see Supplementary Methods and Supplementary Fig. 1). The algorithm combines the Lubachevsky–Stillinger algorithm[37] that consists of standard event-driven molecular dynamics (MD) and slow compression, with Monte Carlo swaps of particle diameters. The MD time is expressed in units of $\sqrt{\beta m \bar{D}^2}$, where the particle mass $m$ and mean diameter $\bar{D}$, as well as the inverse temperature $\beta$, are all set to unity. In other words, a particle travels over a distance of the order of the diameter within a unit MD time. From the thermodynamic point of view, the system is still in the liquid but we work at density $\varphi_g$ sufficiently above the mode-coupling theory (MCT) crossover density $\varphi_d$. Then, once we switch off the particle swapping and return to the natural dynamics simulated by MD, the $\alpha$-relaxation time has become much larger than our MD simulation timescales so that the system behaves essentially as a solid. This glass is thus ultrastable, in a sense similar to those obtained by vapour deposition experiments[38–40]. At a given density $\varphi_g$, we prepare many of such equilibrated configurations that are statistically independent from each other, and we call them samples in the following.

Second, subsequently the equilibrated configuration is compressed up to a target density $\varphi$ with a compression rate $\delta_g = 10^{-3}$. From a single sample that is a starting equilibrated configuration at $\varphi_g$, we generate an ensemble of compressed glasses at $\varphi$, obtained by choosing statistically independent initial particle velocities drawn from the Maxwell–Boltzmann distribution. We call each of such compressed glasses as a realization in the following. These realizations are out of equilibrium, since they no longer follow the liquid equation of state (EOS), but we consider that they remain in restricted equilibrium[29] for $\varphi < \varphi_G$, that is, they are equilibrated within the given glass state determined by the sample. The MD preserves the kinetic energy so that the system remains at the unit temperature throughout our simulations. The typical scale of the vibrations of the particles within the glass states depends on $\varphi$. For instance, it varies from $10^{-1}$ to $10^{-2}$ for $\varphi = 0.645$ to $\varphi = 0.688$ (for $\varphi_g = 0.643$, see Fig. 2 of ref. 34), so that particles make 10 to $10^2$ collisions within a unit MD time.

Third, for a given realization, a simple shear is applied. The simple shear is modelled by an affine deformation of the $x$-coordinates of all particles, $x_i \rightarrow x_i + \gamma z_i$, under the Lees–Edwards boundary condition[41] with fixed system volume. The shear strain is increased quasistatically with a small constant shear rate $\dot{\gamma} = 10^{-4}$, such that the shear rate dependence is

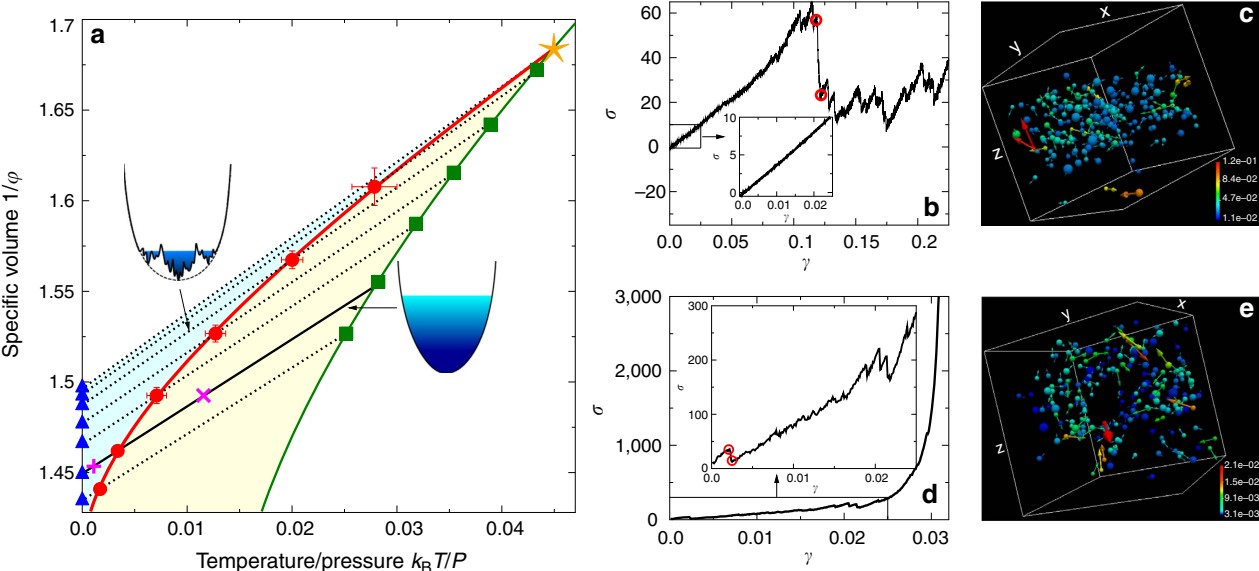

**Figure 1 | Typical stress responses under quasistatic shear.** (**a**) Illustration of the protocol on the polydisperse hard sphere glass phase diagram (adapted from ref. 34), where $k_BT/P = 1/(\rho p)$. The MCT dynamical crossover (yellow star) is located at $\varphi_d = 0.594(1)$ along the equilibrium liquid EOS (green line). Using the swap algorithm we first prepare equilibrium samples at various densities $\varphi_g$ (green squares) whose pressure obeys the Carnahan–Stirling empirical liquid EOS[34]. Next we switch off the swap algorithm, and perform compression annealing from $\varphi_g$ to jamming (blue triangles), producing realizations of compressed glasses at various densities $\varphi$. The system is now out of equilibrium and the pressure follows the glass EOSs $p \propto 1/(\varphi_J - \varphi)$ (black dotted lines)[34]. The Gardner transition $\varphi_G$ (red circles and line) separates the stable (light yellow regime) and the marginally stable (light blue regime) glass phases. The insets show schematic depictions of free-energy landscapes in these two different phases. As an example, an equilibrium configuration is prepared at $\varphi_g = 0.643$, and compressed (solid black line) up to $\varphi_J = 0.690(1)$. We show typical stress–strain curves under quasistatic shear with increasing $\gamma$, using a single realization of the compressed glass of $N = 1,000$ particles, at (**b**) $\varphi = 0.670$ (pink cross) and (**d**) $\varphi = 0.688$ (pink plus) that are below and above $\varphi_G = 0.684(1)$ respectively. Curves in (**b,d**) are zoomed in (insets) for $\gamma \leq 0.025$, to show the different small-$\gamma$ behaviours in the two cases. The real-space vector fields of particle displacements are visualized in (**c**) for a yielding event (between the two red circles in (**b**)), and (**e**) for a MPE (between the two red circles in (**d**)), where each sphere is located at the equilibrium position before yielding/ MPE, and each vector represents the displacement during yielding/MPE. We have subtracted the affine part caused by shear from the displacements, and only show top 20% particles with large displacements. A shear band around the middle of the $z$-axis is observed in (**c**). The sizes of particles are reduced by a factor of 0.4, and the vectors are amplified in length by a factor of 2 in (**c**) and a factor of 15 in (**e**). The colour represents the magnitude of displacement.

negligible in the regime $\varphi < \varphi_G$ (see Supplementary Fig. 6 for a discussion on the $\dot{\gamma}$ dependence), and the shear stress $\Sigma$ is measured at different $\gamma$. The shear stress $\Sigma$ and the pressure $P$ are both calculated from interparticle interactions due to collisions between hard sphere particles. For convenience, we introduce reduced pressure $p = \beta P/\rho$ and reduced stress $\sigma = \beta \Sigma/\rho$, where $\rho$ is the number density of the particles (see Supplementary Note 1). Note that as the pressure, the shear stress is entirely due to momentum exchanges between the particles so that the rigidity is purely entropic in hard sphere systems. Furthermore, because shear stress and pressure have the same physical dimension, it is convenient to introduce a rescaled stress $\tilde{\sigma} = \sigma/p$.

**Breakdown of elasticity.** Figure 1 shows the phase diagram for our polydisperse hard sphere model, and typical stress–strain curves of individual realizations in different density regimes. In the stable glass phase $\varphi_g < \varphi < \varphi_G$ (Fig. 1b), the stress–strain curve shows a smooth linear (harmonic) response regime at small $\gamma$, followed by a sharp drop of the stress $\sigma$, signalling the yielding of the system. At yielding, a system-wide shear band emerges (see Fig. 1c), and the system is driven out of a free-energy metastable glass basin. After yielding, the system enters a steady flow state, similar to those observed in athermal amorphous solids under quasistatic shear[6,42]. In the Gardner phase

$\varphi_G < \varphi < \varphi_J$, where $\varphi_J$ is the jamming density, the harmonic response is punctuated by mesoscopic plastic events (MPEs) that can happen at very small $\gamma$ (see Fig. 1d). These MPEs correspond to sudden avalanche-like heterogeneous rearrangements of particle positions without formation of band-like patterns (see Fig. 1e). Similar MPEs have been observed in quasistatic shear simulations at zero temperatures[2,6], but our simulations are performed at finite temperatures. Note that the details of the plastic events, including the locations of yielding, jamming and MPEs, depend on the samples (see Supplementary Fig. 8) and realizations (see Supplementary Fig. 7). For the behaviour of the stress–strain curves averaged over many realizations and samples, see Supplementary Figs 2–4, as well as Supplementary Notes 2 and 3.

For large $\varphi$, the stress $\sigma$ grows dramatically at large $\gamma$, and appears to diverge (see Fig. 1d). This shear jamming phenomenon is due to the dilatancy effect of hard sphere glasses under shear: the pressure $p$ increases with $\gamma$ when the system volume is fixed. Note that if $p$ is kept as a constant when $\gamma$ is increased, then the volume expands due to the dilatancy effect. In that case, shear jamming does not appear and shear yielding is recovered (see Supplementary Fig. 5). While the switching from shear yielding to shear jamming with increasing $\varphi$ is not a consequence of the Gardner transition, it implies that the system is trapped more deeply in the metastable basin, and that the activated barrier crossing between metastable basins becomes forbidden. However,

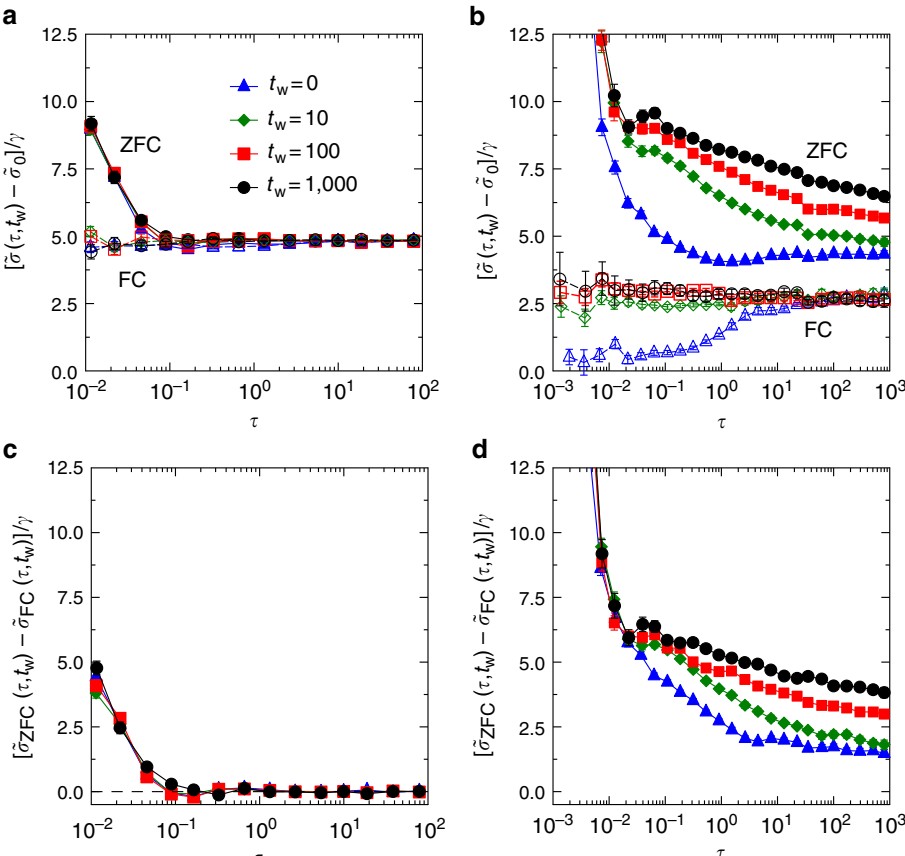

**Figure 2 | Relaxation of shear stress.** Relaxations of the rescaled ZFC shear stress $\tilde{\sigma}_{ZFC} = \sigma_{ZFC}/p$ (filled symbols) and the rescaled FC shear stress $\tilde{\sigma}_{FC} = \sigma_{FC}/p$ (open symbols) show different behaviours at (**a**) $\varphi = 0.670$ and (**b**) $\varphi = 0.688$, corresponding to the pink plus and cross in Fig. 1, respectively (the Gardner transition density $\varphi_G = 0.684(1)$ (ref. 34)). We show results for several different waiting time $t_w$, under an instantaneous increment of shear strain $\gamma = 10^{-3}$. Data are averaged over many realizations of compressed glasses obtained from a single equilibrated sample at $\varphi_g = 0.643$ with $N = 1,000$ particles. Here the rescaled remanent stress $\tilde{\sigma}_0$ is measured in the ZFC protocol at $\varphi$, after the longest waiting time $t_w = 1,000$ and before the shear strain is applied. The difference $\tilde{\sigma}_{ZFC}(\tau, t_w) - \tilde{\sigma}_{FC}(\tau, t_w)$ quickly vanishes and does not show significant $t_w$ dependence at (**c**) $\varphi = 0.670$, while it decays much slower and shows a strong $t_w$-dependent ageing effect at (**d**) $\varphi = 0.688$. Note that by definition, $\tilde{\sigma}_{FC}(t)$ is a one variable function, but we plot it here as $\tilde{\sigma}_{FC}(\tau, t_w)$ to compare it with $\tilde{\sigma}_{ZFC}(\tau, t_w)$. The pressure $p$ is independent of time and protocol, in both cases (see Supplementary Fig. 11). The error bars denote the s.e.m.

the emergence of subbasins in the Gardner phase[28,34] implies that even though the usual relaxation ($\alpha$-relaxation) is frozen, an additional slow dynamics may appear. This aspect is explored below.

**Ageing and slow dynamics.** We next show that in the Gardner phase, the relaxation of shear stress becomes complicated, accompanied by ageing and a slow dynamics. Due to the similarity between the Gardner transition and the spin glass transition, here it is very useful to first recall what happens in spin glasses that are essentially disordered and highly frustrated magnets[43,44]. The mean-field spin glass theory has suggested complex free-energy landscapes of spin glasses manifested as continuous replica symmetry breaking[45], much as what happens in the Gardner phase of hard sphere glasses[27,28]. Remarkably, this feature is predicted to have a reflection in the dynamics, resulting in nontrivial dynamical responses to external magnetic field, and ageing effects in the relaxation of magnetization[46–48]. In experiments, the simplest approach to examine the intriguing features of the dynamics is a combination of the so-called zero-field cooling (zfc) and field cooling (fc) protocols. In the zfc protocol, one cools a spin glass sample from a high

temperature in the paramagnetic phase down to a target temperature $T$, where a magnetic field $h$ is switched on and one measures the increase of the magnetization. In the fc protocol, one first switches on the magnetic field $h$, and then subsequently cools the system down to the target temperature $T$ and measures the remanent magnetization. The key point is that, in the two protocols, the order of cooling and switching on of the magnetic field is reversed. In such experiments[49,50], the magnetizations observed in the zfc/fc protocols are the same if the working temperature $T$ is higher than the spin glass transition temperature, while the fc magnetization becomes larger than the zfc magnetization if $T$ is lower than the spin glass transition temperature. The anomaly, that is, the difference between the zfc and fc magnetizations, is naturally explained by the mean-field theory[45]. Furthermore, examinations of the ageing effects by these protocols give detailed information about the complex free-energy landscape[32,33,46–48].

It has been pointed out theoretically that the shear on structural glasses plays a very similar role as the magnetic field on spin glasses[9,51], and that the relaxation of the shear stress should also reflect the complex free-energy landscape encoded by the continuous replica symmetry breaking solution in the

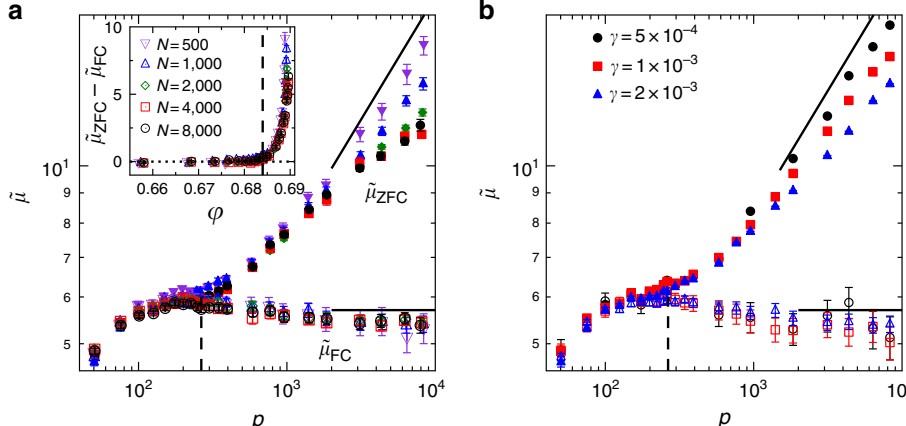

**Figure 3 | Protocol-dependent shear modulus. (a)** The rescaled shear modulus $\tilde{\mu}=\mu/p$, obtained from both ZFC (filled symbols) and FC (open symbols), is plotted as a function of $p$, for $\varphi_g = 0.643$ and several $N$. The data are obtained by using $\gamma = 2 \times 10^{-3}$, and are averaged over $N_s \approx 200$ samples, and $N_r \approx 100$ individual realizations for each sample. The two shear moduli $\mu_{ZFC}$ and $\mu_{FC}$ coincide below $p_G$ (vertical dashed line), and become distinct above, where $p_G = 265$ (ref. 34). The data are compared with the large $p$ scalings predicted by the mean-field theory $\mu_{ZFC} \sim p^{1.41574}$ and $\mu_{FC} \sim p$ (black solid lines). The difference $\mu_{ZFC} - \mu_{FC}$ is plotted as a function of $\varphi$ in the inset, where the vertical dashed line represents $\varphi_G = 0.684$ (ref. 34). **(b)** Rescaled ZFC and FC shear moduli obtained from a few different $\gamma$, for $N = 1,000$ systems. The error bars denote the s.e.m.

Gardner phase (see Supplementary Fig. 9, and Fig. 2 of ref. 10). The shear strain and stress in structural glasses correspond to the magnetic field and magnetization in spin glasses, respectively. Furthermore, apparently compression in hard sphere glasses corresponds to cooling in spin glasses. Therefore, inspired by the zfc/fc experiments in spin glasses, we design two distinct protocols that are combinations of compression and shear exerted in reversed orders. In the zero-field compression (ZFC) protocol, we first compress the configuration from $\varphi_g$ to $\varphi$, and set the time to zero. We then wait for time $t_w$ before a shear strain $\gamma$ is applied instantaneously (see Supplementary Methods), and measure the relaxation of the stress $\sigma_{ZFC}(t, t_w)$ as a function of the time $\tau = t - t_w$ elapsed after switching on the strain. On the other hand, in the field compression (FC) protocol, we first apply an instantaneous increment of shear strain at the initial density $\varphi_g$, compress the configuration to $\varphi$ and set the time to zero. Then, we measure the relaxation of the stress $\sigma_{FC}(t)$ as the function of the elapsed time $t$.

For $\varphi < \varphi_G$, no ageing effect is observed, and the dynamics is fast. The $\sigma_{ZFC}(t, t_w)$ is stationary or time translationally invariant, that is, $\sigma_{ZFC}(\tau, t_w) = \sigma_{ZFC}(\tau)$, depending only on the time difference $\tau = t - t_w$ but not on the waiting time $t_w$ (see Fig. 2). After a timescale $\tau_b$ corresponding to the ballistic motions of particles[34], the ZFC stress $\sigma_{ZFC}(\tau, t_w)$ converges quickly to $\sigma_{FC}(t)$ that is almost a constant in time.

In contrast, for $\varphi > \varphi_G$, $\sigma_{ZFC}(\tau, t_w)$ displays strong $t_w$-dependent ageing effects manifesting the out-of-equilibrium nature of the system, as well as a slow dynamics. In such a situation, different large time limits can emerge depending on the order of $\tau \to \infty$ and $t_w \to \infty$ (ref. 52). An important feature that can be seen in Fig. 2 is that $\sigma_{ZFC}(\tau, t_w)$ exhibits a plateau suggesting the existence of a large time limit $\sigma_{ZFC} \equiv \lim_{\tau \to \infty} \lim_{t_w \to \infty} \sigma_{ZFC}(\tau, t_w)$ where $t_w \to \infty$ is taken before $\tau \to \infty$. On the other hand, $\sigma_{FC}(t)$ is again essentially constant in time $t$ (for $t > \tau_b$) and we shall denote it as $\sigma_{FC}$. In the reversed order of the large time limits, we expect that the ZFC shear stress decays to the FC one, $\lim_{t_w \to \infty} \lim_{\tau \to \infty} \sigma_{ZFC}(\tau, t_w) = \sigma_{FC}$. However, the convergence becomes slower as $t_w$ increases, and its corresponding timescale could be beyond the simulation time window, as shown in the case of Fig. 2.

Apparently $\sigma_{ZFC}$ is larger than $\sigma_{FC}$ when $\varphi > \varphi_G$, implying the ergodicity breaking. The ageing effect and the slowing down of dynamics show the similarities between the Gardner transition and the liquid–glass transition, demonstrating that the Gardner transition could be considered as a 'glass transition within the glass phase' (see also Supplementary Note 3). In a sharp contrast, because the Gardner transition is absent in a crystal, its shear stress relaxes faster when $\varphi$ increases, and no ageing is present.

**Protocol-dependent shear modulus.** The above observation suggests that the linear shear moduli measured by the two protocols should be distinct in the Gardner phase. We determine the apparent shear modulus $\mu$ as $\mu_{ZFC} = (\sigma_{ZFC} - \sigma_0)/\gamma$ and $\mu_{FC} = (\sigma_{FC} - \sigma_0)/\gamma$, where $\sigma_0$ is the remanent shear stress at $\varphi$ before $\gamma$ is applied. The shear strain is increased quasistatically with rate $\dot{\gamma} = 10^{-4}$ up to a predetermined small target $\gamma$. The shear stress is measured at $\tau = 1$ after waiting for $t_w = 10$. Details on the time and $\varphi_g$ dependences of the shear modulus are discussed in Supplementary Figs 10, 13 and 14.

Figure 3 shows that while $\mu_{ZFC}$ and $\mu_{FC}$ are indistinguishable in the stable glass phase $\varphi < \varphi_G$ (or $p < p_G$), they become clearly distinct in the Gardner phase $\varphi > \varphi_G$ (or $p > p_G$). For a similar result of a two-dimensional bidisperse hard disk model, see Supplementary Fig. 16. This behaviour of shear modulus is a consequence of the time dynamics of the shear stress illustrated in Fig. 2: at the timescales used to measure the shear modulus ($\tau = 1$ and $t_w = 10$), the two shear stresses $\sigma_{ZFC}$ and $\sigma_{FC}$ have converged to the same value for $\varphi < \varphi_G$, but remain different for $\varphi > \varphi_G$. The bifurcation point determines the Gardner transition threshold $\varphi_G$ (or $p_G$). Within the numerical accuracy, the $\varphi_G$ determined from this approach is fully consistent with the previous estimate based on particles' vibrational motions and caging order parameters[34]. To further test this result, we perform detailed analysis on its dependence on the number of particles $N$ and the shear strain $\gamma$, as discussed below.

We find no appreciable finite size effects for $\mu_{FC}$ (see Fig. 3a) that is in contrast to the observation in nonequilibrated systems, where $\mu_{FC}$ decreases to zero in the thermodynamic limit[13]. It

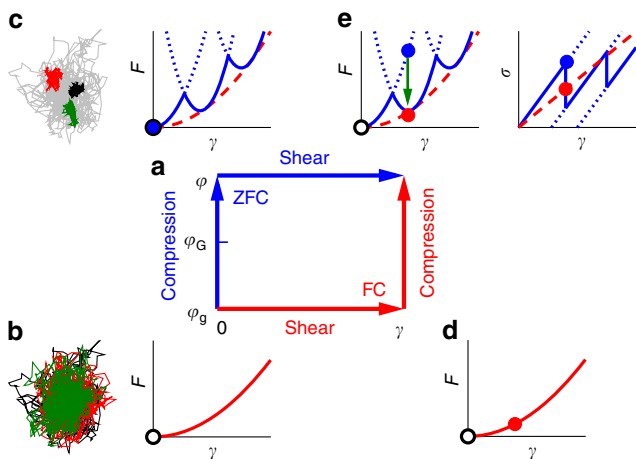

**Figure 4 | Illustration of protocols.** We show the evolution of the free-energy landscape and the state point $(\varphi, \gamma)$ under compression and shear. (**a**) In the ZFC protocol, the system is first compressed and then sheared, while the order is reversed in the FC protocol. (**b**) State point $(\varphi_g, 0)$: the schematic free-energy $F$ as a function of the strain $\gamma$ at the initial density $\varphi = \varphi_g$ before compression. We assume that the initial state point (black open circle) is located at the minimum of the parabola. To show an example of the real-space particle caging, we also plot three independent trajectories of the same tagged particle in the same two-dimensional sample (see Supplementary Note 5). (**c**) State point $(\varphi, 0)$: if the system is compressed first to $\varphi$ (above the Gardner transition density $\varphi_G$), the free-energy basin (red dashed line) splits into many subbasins (blue line): the state point (blue solid circle) becomes trapped in one of the subbasins. The dotted blue lines represent the metastable region of the subbasins. The split of free-energy basin corresponds to the split of cage in the real space (as an example, see the independent trajectories representing three split cages). (**d**) State point $(\varphi_g, \gamma)$: on the other hand, if the system is sheared first, the state point (red solid circle) is forced to climb up the parabola of the basin. (**e**) State point $(\varphi, \gamma)$: after both shear and compression, the state point can be located at different positions in the same free-energy landscape, depending on the order of the compression and shear. In the ZFC case, the state point (blue solid circle) is forced to climb up the subbasin where it is trapped, while it can remain at lower free-energy state in the FC protocol (red solid circle). Because subbasins are metastable (dotted blue line), MPEs occur with increasing $\gamma$ in a quasistatic shear, and slow relaxation occurs for a fixed $\gamma$ (green arrow). The shear stress $\sigma$ is determined by $\sigma \sim dF/d\gamma$ (right panel), and the shear modulus by $\mu = d\sigma/d\gamma \sim d^2F/d\gamma^2$. The stress–strain curves show that for $\varphi > \varphi_G$, $\mu_{ZFC}$ (slope of blue line) is larger than $\mu_{FC}$ (slope of dashed red line).

suggests that preparing deeply equilibrium configurations is the key to observe the nonvanishing $\mu_{FC}$. While the shear moduli measured around the Gardner transition, and therefore the determination of $\varphi_G$, are $N$ independent, stronger finite size effects are observed for $\mu_{ZFC}$ at large $p$ near the jamming limit: $\mu_{ZFC}$ is lower in larger systems, suggesting a stronger nonlinear effect. Nevertheless, the data of $\mu_{ZFC}(p)$, with a fixed $\gamma$, appear to converge for $N \gtrsim 2,000$, confirming that $\mu_{ZFC}(p)$ and $\mu_{FC}(p)$ remain distinguishable in the thermodynamic limit, for $\varphi > \varphi_G$.

Regarding the $\gamma$-dependence, Fig. 3b shows that, within the numerical accuracy, $\mu_{FC}$ is independent of $\gamma$, as long as $\gamma$ is sufficiently small. On the other hand, for $\varphi > \varphi_G$ and a given $N$, $\mu_{ZFC}$ slightly increases with decreasing $\gamma$. This result shows that in the Gardner phase, the nonlinear effect on $\mu_{ZFC}$ remains even for very small $\gamma$, consistent with the observation of elasticity breakdown in Fig. 1. Such nonlinear effects are observed for any $N$ studied (see Supplementary Fig. 12), and we expect that in

the thermodynamic limit $N \rightarrow \infty$, a pure linear behaviour of $\mu_{ZFC}$ can only exist in the limit $\gamma \rightarrow 0$ (ref. 53). The vanishing of the pure elastic regime distinguishes the Gardner phase from the normal glass and crystalline phases. For a more detailed discussion on how the shear moduli depend on the strain $\gamma$, the particle number $N$, the initial density $\varphi_g$ and the waiting time $t_w$, see Supplementary Note 4.

For $\varphi > \varphi_g$, the mean-field theory predicts two power-law scalings in the large $p$ limit[10]: $\mu_{ZFC} \sim p^\kappa$ with $\kappa = 1.41574...$, and $\mu_{FC} \sim p$. The first scaling has also been derived semiempirically by an independent approach[54]. We find good agreement between the theory and simulation on the scaling of $\mu_{FC}$ (see Fig. 3). For $\mu_{ZFC}$, a noticeable discrepancy is observed in the limit of large $N$ for a fixed finite $\gamma$ (Fig. 3a), but the discrepancy decreases when $\gamma \rightarrow 0$ for a fixed $N$ (Fig. 3b), or when $N$ is decreased for a fixed $\gamma$ (Fig. 3a). This is because the mean-field $\mu_{ZFC}$ is obtained in the pure linear response limit $\gamma \rightarrow 0$, while the nonlinear effect caused by MPEs would increase with $\gamma$ and $N$, as discussed above. The scaling $\mu_{FC} \sim p$ is consistent with the experimental observation in emulsions[21,22]. Considering the experimental system is possibly not deeply equilibrated, we expect that the relaxation of experimental $\mu(t)$ is sufficiently fast, and the measurement was performed in the long time limit $\mu(t \rightarrow \infty) \rightarrow \mu_{FC}$ (see the discussion of Fig. 2).

**Interpretation of results.** The Gardner transition is a consequence of the split of glass basins in the phase space[28], and the split of particle cages in the real space (see Fig. 4). The schematic plot of the free-energy $F$ as a function of $\gamma$ in Fig. 4 illustrates how a glass basin splits into many subbasins once the system is compressed above $\varphi_G$. Here we interpret our results based on this free-energy landscape viewpoint. First, in the ZFC protocol, the system intends to remain in one of the subbasins after compression (note that different realizations may end up in different subbasins), but as $\gamma$ increases in a quasistatic shear procedure (Fig. 1), it may become unstable where the shear stress drops abruptly, resulting in a MPE. The MPE could be interpreted as shear-induced barrier crossing between subbasins, analogous to the barrier crossing between basins in a yielding event. Second, if $\gamma$ is fixed, the shear stress relaxes with time, and according to the Arrhenius law, the emergence of barriers between subbasins would result in a slowing down of the relaxation dynamics with $\varphi$ (Fig. 2). The appearance of ageing further reveals the emergence of complex structures within a basin, similar to the mechanism of ageing in the glass transition[52]. Third, because in the FC protocol, the system can overcome the subbasin barriers, the $\mu_{FC}$ always corresponds to the second-order curvature of basins, rather than that of subbasins as in the $\mu_{ZFC}$ case. This results in $\mu_{FC} < \mu_{ZFC}$ for the regime $\varphi > \varphi_G$ as observed in Fig. 3. Note that according to Fig. 4e, one should obtain a shear modulus close to $\mu_{ZFC}$ if an additional strain is applied after FC, as confirmed in Supplementary Fig. 15. On the other hand, no basin split occurs and therefore two protocols are equivalent in the stable phase $\varphi < \varphi_G$. A previous study[13] has shown that $\mu_{ZFC} = \mu_{FC}$ for crystals. The similarity between crystals and stable glasses further confirms that their free-energy basins are similarly structureless.

**Discussion**
We wish to stress that our data cannot exclude that the Gardner transition becomes just a crossover in finite dimensions, such that no real phase transition exists. Yet, irrespective of the sharpness of the Gardner transition, we rationalize here in a unified framework all the observations obtained on the rheological

behaviour of the simple hard sphere glass, and find quantitatively reasonable agreement between the theory and simulations. Thus, even if the Gardner transition is not sharp in the thermodynamic limit, for accessible sizes in numerical simulations, and likely for those in experiments as well[35], a behaviour reminiscent of the transition can be clearly observed.

Finally, we make remarks on experimental consequences. It is an intriguing question to clarify whether the phase diagram presented in Fig. 1a is generic in a wide range of amorphous solids, ranging from different kinds of glasses to soft matter such as colloids (one can choose to change the temperature or pressure as the control parameter depending on specific systems). The crucial point is to keep track of the dynamical effects that might have been overlooked in some previous experiments, for the following two reasons. First, in reasonably stabilized dense systems, the liquid EOS (green line in Fig. 1a) and the Gardner line (red line) becomes separated enough, so that the liquid dynamics ($\alpha$-relaxation) and the intriguing internal glassy dynamics ($\beta$-relaxation induced by the Gardner transition) can be well separated in timescales. In this respect, recently developed experimental techniques, such as the vapour deposition[38–40] and the high pressure path[55], or the use of sufficiently old natural glasses[56], would provide ideal settings. If such an ideal setting is not possible, one could freeze the $\alpha$-relaxation out of the experimental time window, by working at sufficiently low temperatures or high densities. The second reason is that by experimentally studying the ageing effects due to the internal dynamics of the amorphous solids, the complexity of the free-energy landscape could become manifested as we demonstrated in the present paper.

**Data availability**. The data that support the findings of this study are available in Osaka University Knowledge Archive (OUKA) with the identifier. http://hdl.handle.net/11094/59688.

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

## Acknowledgements

We thank Giulio Biroli, Ludovic Berthier, Patrick Charbonneau, Olivier Dauchot, Anaël Lematre, Corrado Rainone and Pierfrancesco Urbani for useful discussions, and especially Francesco Zamponi, Daijyu Nakayama, and Satoshi Okamura for many stimulating interactions. This work was supported by KAKENHI (No. 25103005 'Fluctuation & Structure') from MEXT, Japan. We also wish to thank financial support from Cybermedia Center, Osaka University. The computations were performed using Research Center for Computational Science, Okazaki, Japan, and the computing facilities in the Cybermedia Center, Osaka Univerisity.

## Author contributions

Both authors contributed to all aspects of this work.

## Additional information

**Competing interests:** The authors declare no competing financial interests.

**Publisher's note**: 

