## [Peer Review File · Nature Communications]

Reviewers' comments:

Reviewer #1 (Remarks to the Author):

The manuscript presents results of numerical simulations of the rheological properties of 3D Hard Spheres glasses. It shows that under quasi-static shear at finite temperatures, there is a transition (or a cross-over) where elasticity breaks down, stress relaxation exhibits slow and aging dynamics, and the apparent shear modulus becomes protocol-dependent. These results are clearly interesting both from the theoretical point of view, since this cross-over could be related to the Gardner transition found in mean-field analysis, and from the experimental point of view since it clearly points toward experimental protocols to study similar questions.

I think the results are important and very intriguing, they confirm (at least to a certain extent) previous theoretical predictions and at the same time will certainly trigger new research directions.

Since the paper is also well written, I recommend it for publication in its present form.

Reviewer #2 (Remarks to the Author):

In this paper the authors investigate the fate of dense hard sphere glasses at high pressure. In a series of recent works it has been claimed that the mean field theory of the glass phase predicts a glass-to-glass transition, the Gardner transition, where glasses enter into a so called marginally stable phase.

The theoretical predictions have triggered a series of numerical and experimental works. Furthermore the consequences of the Gardner transition on the rheological properties of amorphous solids have been investigated in [Ref. 14 and 15 of the cited references] predicting a breakdown of the elastic behavior at this critical point.

The present paper represents a further investigation of the Gardner transition within the same stream of ideas.

The authors show through smart numerical simulations that the elastic behavior of amorphous solids breaks down deep in the glass phase at a previously identified Gardner point confirming the theoretical predictions.

The Gardner transition seems to be a kind of spin glass transition.

In spin glasses the response of the system to an external magnetic field depends on the protocol with which the system is prepared and perturbed. Two kinds of protocols have been investigated: zero field cooling (ZFC) and field cooling (FC). Once the transition point is crossed, the two relative magnetic responses become different.

The authors show that the same happens in a simple glass former.

In this case a natural way to define responses is by applying a strain perturbation that plays the role of the external field.

Thus one can define zero field compression and field compression protocols in analogy with spin glasses (this justifies also the use of the same acronyms FC and ZFC).

The authors show that at a previously identified Gardner transition point, both the stress and the shear modulus depend on the perturbation protocol.

Furthermore, the numerical simulations show strong aging effects once the Gardner transition is crossed, as was previously shown in Ref. 28 of the cited references.

Thus, the present paper suggests that the anomalies observed in the elastic behavior of amorphous solids deep in the glass phase can be the signature of a new phase transition.

It will surely trigger new investigations on the relation between soft glassy rheology and the physics of what the authors call marginally stable glasses.

Some detailed remarks are listed below.

1) It seems that the definition of the yielding strain γ_Y in the main text is different from the one in the supplementary material. In one case it is the strain at the peak of the stress strain curve, in the other it is the big jump after the peak.

2) Can the author explain why the small systems obey better to the mean field slope predicted for μ_{ZFC} in Fig. 3a?

Do they have a simple explanation for that?

3) I think that the connection between structural glasses and spin glasses should be underlined and the relevant literature for magnetic susceptibilities in the spin glass case could be cited.

4) I do not understand the name "micro-yielding". There is still no complete understanding of the yielding transition itself and the name micro-yielding is confusing to me.

I would suggest to call the avalanches that appear along the stress-strain curve "plastic events".

5) I do not understand the precise meaning of the sentence "After yielding, it becomes a fluid, such that different basins are explored freely". What does it mean that the basins are explored freely? Do the author have a precise picture for the flow beyond yielding?

Reviewer #3 (Remarks to the Author):

The authors of the submitted manuscript propose the use of standard rheological techniques as a tool to investigate the free energy landscape of hard-sphere glass under quasi-static shear at finite temperatures. The mechanical and thermodynamic behavior of amorphous solids under external perturbation, as change of pressure or shear stress, is of large interest for the scientific community due to the very different performance that amorphous materials show compared to crystals. Although many progresses have been achieved in the understanding of this fascinating problem, included the most recent literature, the whole picture is still lacking and under debate, especially for finite temperatures and finite dimensional cases.

The present work aims to perform extensive numerical simulations of a polydisperse mixture of hard sphere whose diameter are distributed according to the probability distribution $P(D) \sim D^{-3}$. Through different numerical experiments, the authors show that - above the Gardner transition - three main phenomena appear and signal the existence of non-trivial behaviors: the elasticity breaks down, the time relaxation of the shear stress shows aging, and the shear modulus becomes protocol-dependent. Asymptotic behavior are then compared to mean-field theoretical results.

I acknowledge the numerical efforts in reproducing such results I think the paper is suitable for publication in Nature Communications. However, I believe that the authors should strengthen the paper following these ideas:

On a general ground, the novelty of the current contribution is not so evident. The manuscript

presents a series of numerical results about amorphous materials which, though in different form, are already acknowledged by the community and therefore the presented simulations do not lend themselves to enough originality in the present form. Author should work more to highlight the original aspects of the present work.

The protocol dependence of the shear modulus reported in Fig. 3 is a consequence of the different time dynamics of the shear stress illustrated in Fig. 2, by the relation that links μ to σ in the two different protocols ZFC and FC. No mention is made about the relation between these two aspects and, although the results are strongly related, they are presented as two independent facts and, furthermore, one additional to the other. A clear presentation would point more distinctly these kind of physical connections.

To reproduce the results illustrated in Fig. 3 it is not clear why the shear stress is measured at $\tau=1$ and after a waiting time for $t_w = 10$. Since the results of the protocol-dependence of the shear modulus depends on the waiting time (as shown in Fig. 2 for the relaxation of the ZFC shear stress) one naturally wonders if the bifurcation of the μ in the two protocols would remain of the same kind and appear at the same value of φ for different waiting times.

Furthermore, the authors find no appreciable finite size effect for the quantity μ_{FC} , which is in contrast to existing observations. They justify this outcome as dependent on the preparation of the equilibrium conditions. To my opinion this point would deserve further explanations and clarifications and it is not well explained in the manuscript.

On the presentation point of view, the manuscript contains some inadequacies that makes it not of enough high quality for Nature Communication. Starting from the Abstract, two full stops are missing at the end of two sentences. Fig. 4 is cited in the main text before Fig. 2 and 3. The caption in Fig. 4 is not adequately well-written and the reader struggles to understanding the reference to all the subfigures. In particular the central figure is never quoted in the caption, as well as the trajectories on the top-left and the second figure from the top right (regarding this, a green arrow is mentioned in the caption but with no reference to a specific panel in the picture). The quantity φ_J in the main text at line 78 is only introduced in the caption of Fig. 1 and never before in the text. Other inaccuracies are present in the main text and figure captions. Authors should pay more attention to these details if they aspire to produce a high quality paper for Nature Comm.

Reviewer #4 (Remarks to the Author):

The paper starts from the assumption of the existence of a Gardner transition in hard sphere glasses, which has been theoretically predicted at the mean field level (infinite dimensions). The major claim is that this transition has signatures in the rheological properties, which can then be used to reveal the existence of the transition. Thus, the paper set-up a numerical model in finite (3) dimensions, prepares configurations on both sides of the transition, and investigates their rheological properties. It finds differences. The importance of this result is not well emphasized in the manuscript. My guess is that the authors, as only mentioned in the conclusions, expect their approach to be of potential use in experiments, as related to routine measures of the stress. Indeed, other approaches to reveal signature of the transition requires the investigation of the trajectories of the particles, which in general are not easily accessed in experiments.

The paper is not conceptually novel. A related work (ref. 35) PRL 117, 228001 (2016), focuses on the same problem from an experimental point of view. It considers a different system (granular), and different way of exploring the phase space (vibrations rather than shearing). Yet the result is

conceptually very similar, and there are strong overlaps. In addition, the present work appears to be the numerical extension of Ref. 14, where the effect of the Gardner transition on the rheological properties is discussed. In Ref. 14 the jerky versus smooth stress-strain relations are associated to the different phases, as in the present manuscript. Possibly, what is new here is the numerical investigation of the aging of the stress. Since this paper is mainly confirming previous results and speculations, I do not think it will influence thinking in the field.

The results provided by the authors are certainly convincing, also considering that they are in line with previous results and theoretical speculations. The numerical approach and the statistical analysis are well detailed in the SI information.

As a final remark, I notice that the exposition is somehow involved, and that the manuscript is not able to smoothly convey its results. This is particularly true for the section describing Fig. 2.

Regardless of the editorial decision on this manuscript, I suggest the authors to review and simplify the exposition to better convey their results.

From the technical side:

- I do have difficulties understanding why the authors decided to focus on a system of $N = 1000$ particles with diameters distributed according to a power law. Of course, $N = 1000$ is too small for any definition of power law to make sense. I understand that the authors are following previous works, but this is not a good reason for this choice. Perhaps the authors are aware of this point, as at the end of the SI they introduced a section discussing results on a bidisperse two-dimensional system of hard disks.

- What is the reason why different configurations at volume fraction ψ are obtained compressing the same equilibrium configuration, just changing the initial velocities? Since the equilibrium state is well below the MC density, the system is not going to relax and explore its phase space during compression. Thus, the different configurations at volume fraction ψ the paper average over are expected to be extremely correlated.

- What is the criterion used to define the yielding transition? The circles in Fig.1 b and c appear to be placed at arbitrary point. For example, in pane b one could speculate the yielding transition to occur at $\gamma \sim 0.11$.

REVIEWERS' COMMENTS:

Reviewer #3 (Remarks to the Author):

The authors have adequately answered my critiques.

Reviewer #4 (Remarks to the Author):

The revised version of manuscript NCOMMS-16-25423A better clarified that the manuscript aims at revealing signatures of Gardner transition, which has been predicted to occur at the mean field level, in finite dimensional systems. In particular the manuscript main claim is that the devised numerical protocol used to detect these signatures could be experimentally used.

I am not convinced by this claim, considering that the devised protocol involves the compression of hard-sphere systems, the authors thermalize above the mode-coupling transition using a particle swap algorithm. In order to use this algorithm, the authors are also forced to consider a peculiar system, for which the particle size has a power-law distribution. Experimentally, it is just impossible to thermalize hard-sphere colloidal systems above their mode coupling transition. Therefore, the experimental relevance of these results remains unclear. I'd like also to note that, while the authors stress that their protocol is suitable "to investigate a large variety of experimental systems", they actually only investigate hard-sphere systems. In particular, they investigate hard sphere system close to their jamming transition. Since close the jamming transition these systems are known to have very peculiar properties, e.g. a diverging correlation lengths, an enormous non-affine response, etc., it is quite possible that the observed phenomenology is peculiar to systems close to jamming. Thus, the relevance of these results to other systems is not obvious.

Apart from that, as previously mentioned and now better clarified by the authors, the reported results are not particularly novel, as the role of the Gardner transition in the rheological properties of hard-sphere systems has been previously discussed.

Thus, while I acknowledge the great numerical effort, it seems to me that the manuscript is neither bringing a novel conceptual understanding of the rheological properties of these systems, nor introducing a truly general and simple protocol to investigate these properties. The relevance of this paper outside the community investigating the Gardner transition in hard-sphere systems is therefore unclear.

The authors have clarified my previous technical concerns, and I do only have few minor suggestions:

- The authors should better explain in the manuscript what it means to shear "quasi-statically" with a constant shear rate > 0 . By definition, quasistatically implies zero shear rate. It is a matter of comparing timescales, and the authors should be clear on this point.

- Fig. 1 highlights the difference in the rheological properties of systems prepared outside and within the Gardner phase. Unfortunately, the figure compares states that have both a different density and a different pressure. Thus, it is not clear from the figure whereas the observed differences are due to the Gardner transition, or just to the density/pressure. This point is somehow clarified in the Supp. Material, where the authors show stress-strain curves for different values of the control parameters. I would suggest reviewing this figure to compare states having same pressure or volume fraction.

Answer to the referee comments

Dear Editors:

We wish to warmly thank the referees for their insightful comments. We are also grateful for the suggestions for improving the manuscript. We have closely followed them. The details are provided below and highlighted in red in the accompanying revised manuscript and supplementary material. We hope that the result will be considered appropriate for publication.

Sincerely,
the authors.

Report of Reviewer #1

The manuscript presents results of numerical simulations of the rheological properties of 3D Hard Spheres glasses. It shows that under quasi-static shear at finite temperatures, there is a transition (or a cross-over) where elasticity breaks down, stress relaxation exhibits slow and aging dynamics, and the apparent shear modulus becomes protocol-dependent. These results are clearly interesting both from the theoretical point of view, since this cross-over could be related to the Gardner transition found in mean-field analysis, and from the experimental point of view since it clearly points toward experimental protocols to study similar questions.

I think the results are important and very intriguing, they confirm (at least to a certain extent) previous theoretical predictions and at the same time will certainly trigger new research directions.

Since the paper is also well written, I recommend it for publication in its present form.

We wish to thank the reviewer for the very positive comments.

Report of Reviewer #2

In this paper the authors investigate the fate of dense hard sphere glasses at high pressure. In a series of recent works it has been claimed that the mean field theory of the glass phase predicts a glass-to-glass transition, the Gardner transition, where glasses enter into a so called marginally stable phase. The theoretical predictions have triggered a series of numerical and experimental works. Furthermore the consequences of the Gardner transition on the rheological properties of amorphous solids have been investigated in [Ref. 14 and 15 of the cited references] predicting a breakdown of the elastic behavior at this critical point. The present paper represents a further investigation of the Gardner transition within the same stream of ideas. The authors show through smart numerical simulations that the elastic behavior of amorphous solids breaks down deep in the glass phase at a previously identified Gardner point confirming the theoretical predictions. The Gardner transition seems to be a kind of spin glass transition. In spin glasses the response of the system to an external magnetic field depends on the protocol with which the system is prepared and perturbed. Two kinds of protocols have been investigated: zero field cooling (ZFC) and field cooling (FC). Once the transition point is crossed, the two relative magnetic responses become different. The authors show that the same happens in a simple glass former. In this case a natural way to define responses is by applying a strain perturbation that plays the role of the external field. Thus one can define zero field compression and field compression protocols in analogy with spin glasses (this justifies also the use of the same acronyms FC and ZFC). The authors show that at a previously identified Gardner transition point, both the stress and the shear modulus depend on the perturbation protocol. Furthermore, the numerical simulations show strong aging effects once the Gardner transition is crossed, as was previously shown in Ref. 28 of the cited references. Thus, the present paper suggests that the anomalies observed in the elastic behavior of amorphous solids deep in the glass phase can be the signature of a new phase transition. It will surely trigger new investigations on the relation between soft glassy rheology and the physics of what the authors call marginally stable glasses.

We wish to thank the reviewer for the very positive comments.

Some detailed remarks are listed below.

1) It seems that the definition of the yielding strain γ_y in the main text is different from the one in the supplementary material. In one case it is the strain at the peak of the stress strain curve, in the other it is the big jump after the peak.

We define the yielding strain γ_y as the strain at the peak of the stress-strain curve, averaged over many samples and realizations (see Supplementary Fig. S2). For the stress-strain curve of a single realization from a single sample, the definition of γ_y is more ambiguous, and in this study we do not attempt to precisely determine γ_y for each single stress-strain curve. This point has been clarified in the main text and Supplementary Note 2.

The point of Fig. 1b and 1c is to show, phenomenologically and qualitatively, how particles move across yielding. For this purpose, we choose two configurations before and after the yielding respectively (red circles in Fig. 1b). The choice of the locations of the two circles is somehow arbitrary: as long as they are not too far away from the big jump, the shear band in Fig. 1c nevertheless can be recognized, which is the key feature that we want to show here.

2) *Can the author explain why the small systems obey better to the mean field slope predicted for μ_{ZFC} in Fig. 3a? Do they have a simple explanation for that?*

We thank the reviewer for raising this point. Here we provide a simple explanation. First of all let us note that the theoretical analysis in Ref. [10] is done in the linear response limit $\mu = \lim_{\gamma \rightarrow 0} (\sigma/\gamma)$, and that plastic events which may happen at finite γ can reduce the shear stress with respect to what one expects assuming the linear response $\sigma = \mu\gamma$. Now let us clarify how μ_{ZFC} depends on strain γ and the system size N , near the jamming limit.

First, for a given γ , smaller systems obey better with the mean-field prediction as shown in Fig. 3a. This is because that smaller systems have fewer mesoscopic plastic events, or MPEs (what we called *micro-yielding* in the old version). Indeed, in Ref. [53], the authors measured numerically the mean strain $\delta\gamma_1$ at which the first MPE takes place in amorphous solids, and found a finite-size scaling $\delta\gamma_1 \sim N^\beta$ with $\beta \approx -0.62$. This result suggests that, in larger systems, MPEs are easier to occur, which reduce the measured μ_{ZFC} . As $N \rightarrow \infty$, MPEs become unavoidable at any finite shear strain as $\delta\gamma_1 \rightarrow 0$. In other word, for a sufficiently small system, if its $\delta\gamma_1$ is larger than the finite γ used in our measurement, then MPEs become rare and the system behaves like an elastic solid. Considering that the μ_{ZFC} predicted by the mean-field theory is only concerned about the linear response limit, one should expect that the data with smaller N (with fixed γ) or smaller γ (with fixed N) should agree better with the theoretical prediction.

Second, for any given N , μ_{ZFC} increases with decreasing γ . Our additional data in Supplementary Fig. S12 show that this is true for any N studied. We thus do not exclude the following possibility: for any given N , the scaling would agree with the mean-field prediction as $\gamma \rightarrow 0$. This conjecture is partially supported by the fact that the mean-field theoretical jamming exponents, which characterize the critical distribution of small inter-particle gaps and weak contact forces, are accurately consistent with simulations in finite dimensions (Supplementary Ref. [11]). However, delicate treatments are required for this purpose. Based on our present data, we cannot draw a conclusion.

We have included this discussion in the modified manuscript (see the section *Protocol-dependent shear modulus* in the main text, and sections *Dependence on the shear strain γ* and *Dependence on the number of particles N* in Supplementary Note 4.

3) *I think that the connection between structural glasses and spin glasses should be underlined and the relevant literature for magnetic susceptibilities in the spin glass case could be cited.*

We thank the reviewer very much for the suggestion. We have included a discussion on the connection to the spin glass systems in the revised manuscript. Relevant references on spin glasses are cited. See the section *Aging and slow dynamics*. We also put a remark on the connection to spin glasses in the 3rd paragraph of the introduction in the revised manuscript.

4) *I do not understand the name “micro-yielding”. There is still no complete understanding of the yielding transition itself and the name micro-yielding is confusing to me. I would suggest to call the avalanches that appear along the stress-strain curve “plastic events”.*

To avoid the confusion, we have changed *micro-yielding* to *mesoscopic plastic event* (MPE). The word *mesoscopic* is used here to distinguish these plastic events from (i) macroscopic ones, i.e., yielding events, which takes place at a system-wide scale (see Fig. 1c), and (ii) microscopic single bond breaking between interacting particles, because MPEs are collective as shown in Fig. 1e.

5) *I do not understand the precise meaning of the sentence “After yielding, it becomes a fluid, such that different basins are explored freely”. What does it mean that the basins are explored freely? Do the author have a precise picture for the flow beyond yielding?*

Beyond yielding, the system enters a steady flow state in the sense that the mean stress is stationary with increasing strain under *quasistatic* shear. Alternatively one may first consider the usual stationary states of a driven system

under constant shear rate $\dot{\gamma}$ and subsequently take the limit $\dot{\gamma} \rightarrow 0$. In this dynamic view point, the yield stress is the stress which remains in the $\dot{\gamma} \rightarrow 0$ limit. The physical picture of such a quasi-static flow state is discussed in detail in several previous studies, for example Ref. [6]. Although Ref. [6] considers an athermal system, we expect that the physical pictures are similar, giving the fact that the observed stress-strain curves have similar behaviors in the steady state (see Fig. 1b and Fig. 2 in [6]): they are formed by smooth, roughly linear elastic segments interrupted by the discrete jumps representing plastic events. In the energy/free energy landscape picture, the smooth segments correspond to reversible, elastic changes of a particular energy/free energy minimum (basin); the jumps correspond to the shear induced annihilation of that minimum with a barrier.

Thanks to the reviewer's comment, we realize that our previous statement is not precise. We have changed it to "After yielding, the system enters a steady flow state, similar to those observed in athermal amorphous solids under quasi-static shear [6,42]". Because the steady flow state itself is very complicated, and it is not the focus of the present manuscript, we do not intend to expand the discussion here. Instead, we refer the readers to other more detailed studies by providing references.

Report of Reviewer #3

The authors of the submitted manuscript propose the use of standard rheological techniques as a tool to investigate the free energy landscape of hard-sphere glass under quasi-static shear at finite temperatures. The mechanical and thermodynamic behavior of amorphous solids under external perturbation, as change of pressure or shear stress, is of large interest for the scientific community due to the very different performance that amorphous materials show compared to crystals. Although many progresses have been achieved in the understanding of this fascinating problem, included the most recent literature, the whole picture is still lacking and under debate, especially for finite temperatures and finite dimensional cases.

The present work aims to perform extensive numerical simulations of a polydisperse mixture of hard sphere whose diameter are distributed according to the probability distribution $P(D) \sim D^{-3}$. Through different numerical experiments, the authors show that - above the Gardner transition - three main phenomena appear and signal the existence of non-trivial behaviors: the elasticity breaks down, the time relaxation of the shear stress shows aging, and the shear modulus becomes protocol-dependent. Asymptotic behavior are then compared to mean-field theoretical results.

I acknowledge the numerical efforts in reproducing such results I think the paper is suitable for publication in Nature Communications.

We wish to thank the reviewer for the very positive comments.

However, I believe that the authors should strengthen the paper following these ideas:

On a general ground, the novelty of the current contribution is not so evident. The manuscript presents a series of numerical results about amorphous materials which, though in different form, are already acknowledged by the community and therefore the presented simulations do not lend themselves to enough originality in the present form. Author should work more to highlight the original aspects of the present work.

We thank the reviewer for this suggestion. The present study has three key novelties: (i) Through extensive simulations, we examine previous mean-field theoretical predictions on the rheology of hard sphere glasses [10,14], developed in infinite dimensional limits, in two- and three-dimensional systems. Such a task is highly non-trivial since the relation between the infinite dimensional theory and real systems in physical dimensions is unclear. The exploration of this connection has been an active and challenging topic in the fields of spin and structural glasses. (ii) We propose a simple protocol that can be straightforwardly realized in standard shear experiments. Our protocol bypasses the limitations in the previous approach [34, 35], and is suitable for a large variety of experimental systems. We thus believe that our work has a potential to make the examination of the Gardner physics accessible to general amorphous systems in laboratories. (iii) This is the first numerical study about the influence of the Gardner transition on the rheology of amorphous solids.

We have improved the introduction of our manuscript to highlight our novelties. Please see also our answers to related questions raised by Reviewer #4 (first two points).

The protocol dependence of the shear modulus reported in Fig. 3 is a consequence of the different time dynamics of the shear stress illustrated in Fig. 2, by the relation that links μ to σ in the two different protocols ZFC and FC. No mention is made about the relation between these two aspects and, although the results are strongly related, they are presented as two independent facts and, furthermore, one additional to the other. A clear presentation would point

more distinctly these kind of physical connections.

We thank the reviewer for pointing out this weakness. The protocol dependence of the shear modulus reported in Fig. 3 is indeed a natural consequence of the different time dynamics of the shear stress illustrate in Fig. 2. In the modified version, we have explicitly clarified their relations in the second paragraph of the section *Protocol-dependent shear modulus*.

To reproduce the results illustrated in Fig. 3 it is not clear why the shear stress is measured at $\tau = 1$ and after a waiting time for $t_w = 10$. Since the results of the protocol-dependence of the shear modulus depends on the waiting time (as shown in Fig. 2 for the relaxation of the ZFC shear stress) one naturally wonders if the bifurcation of the μ in the two protocols would remain of the same kind and appear at the same value of φ for different waiting times.

Thanks to the reviewer's comment, in the modified manuscript, we have carefully examined the waiting time dependence. Our additional data (Supplementary Fig. S14) confirms that, within our numerical accuracy, the bifurcation of the shear moduli obtained by the two protocols is indeed independent of the waiting time. We have also added a discussion on the waiting time dependence in the section *Dependence on the waiting time t_w* in Supplementary Note 4.

Furthermore, the authors find no appreciable finite size effect for the quantity μ_{FC} , which is in contrast to existing observations. They justify this outcome as dependent on the preparation of the equilibrium conditions. To my opinion this point would deserve further explanations and clarifications and it is not well explained in the manuscript.

We thank the reviewer for pointing out this weakness. It was observed in Ref. [13] that $\mu_{FC} \sim N^{-0.25}$, which suggests that $\mu_{FC} \rightarrow 0$ as $N \rightarrow \infty$. We argue that this difference arises because the systems prepared there are far away from equilibrium. Indeed, previous simulation data show that such a non-equilibrium system can quickly relax to a liquid (see Ref. [18]), whose shear modulus is zero. We thus expect that the N -dependence on μ_{FC} , in particular the vanishing of μ_{FC} in the thermodynamic limit, is due to the fact that the systems are not well equilibrated in Ref. [13]. To clarify this point, we have added a discussion in the section *Dependence on the number of particles N* in Supplementary Note 4.

On the presentation point of view, the manuscript contains some inadequacies that makes it not of enough high quality for Nature Communication. Starting from the Abstract, two full stops are missing at the end of two sentences. Fig. 4 is cited in the main text before Fig. 2 and 3. The caption in Fig. 4 is not adequately well-written and the reader struggles to understanding the reference to all the subfigures. In particular the central figure is never quoted in the caption, as well as the trajectories on the top-left and the second figure from the top right (regarding this, a green arrow is mentioned in the caption but with no reference to a specific panel in the picture). The quantity φ_J in the main text at line 78 is only introduced in the caption of Fig. 1 and never before in the text. Other inaccuracies are present in the main text and figure captions. Authors should pay more attention to these details if they aspire to produce a high quality paper for Nature Comm.

We sincerely appreciate the reviewer for pointing out these inadequacies. We have corrected all of them in the revised version. We have also carefully checked the rest of the manuscript.

Report of Reviewer #4

The paper starts from the assumption of the existence of a Gardner transition in hard sphere glasses, which has been theoretically predicted at the mean field level (infinite dimensions). The major claim is that this transition has signatures in the rheological properties, which can then be used to reveal the existence of the transition. Thus, the paper set-up a numerical model in finite (3) dimensions, prepares configurations on both sides of the transition, and investigates their rheological properties. It finds differences. The importance of this result is not well emphasized in the manuscript. My guess is that the authors, as only mentioned in the conclusions, expect their approach to be of potential use in experiments, as related to routine measures of the stress. Indeed, other approaches to reveal signature of the transition requires the investigation of the trajectories of the particles, which in general are not easily accessed in experiments.

We agree with the reviewer that the key importance of our work is the direct numerical observation of the non-trivial rheological consequences of the complex free-energy landscape, based on a combination of standard protocols which can be potentially implemented in real experiments. We wish to emphasize that the focus of the present study is not to reveal evidence of the Gardner transition, as Refs. [34, 35], but to show that the anomalous rheological

behavior of amorphous solids can be explained by using the concept of Gardner transition. This topic is completely new, and distinguishes our study from the existing approaches mentioned by the reviewer, which are not concerned about rheology.

In the introduction of the revised version, we have emphasized both the significance of our own direct rheological observation, and implications for future experiments in a broader range of glassy systems.

The paper is not conceptually novel. A related work (ref. 35) PRL 117, 228001 (2016), focuses on the same problem from an experimental point of view. It considers a different system (granular), and different way of exploring the phase space (vibrations rather than shearing). Yet the result is conceptually very similar, and there are strong overlaps. In addition, the present work appears to be the numerical extension of Ref. 14, where the effect of the Gardner transition on the rheological properties is discussed. In Ref. 14 the jerky versus smooth stress-strain relations are associated to the different phases, as in the present manuscript. Possibly, what is new here is the numerical investigation of the aging of the stress. Since this paper is mainly confirming previous results and speculations, I do not think it will influence thinking in the field.

The conceptual novelty of our work is that the classical elastic description in solid state physics breaks down for amorphous solids in the Gardner phase. This idea was proposed very recently in a few theoretical studies based on mean-field calculations [10,14], as pointed out by the reviewer. However, it is highly debated that to what extent the mean-field theory is relevant to real systems in two or three dimensions. Here, based on extensive numerical simulations, we provide a clear answer: the mean-field description of the rheological properties of amorphous solids is consistent with numerical observations, for accessible system sizes in simulations. Importantly, we show this consistency not only qualitatively, but also quantitatively, by comparing the mean-field scalings to simulation data. Note that, in general, examining mean-field theories in finite-dimensional systems is a highly non-trivial task. For example, even after more than 30 years of effort, the finite-dimensional persistence of the de Almeida-Thouless transition in spin-glasses in a field, which is akin to the Gardner transition here, is still under debate. Here we certainly do not attempt to make a conclusion on the sharpness of the Gardner transition in finite dimensions, but rather, we show that irrespective of the sharpness, it has an important impact on the rheological behaviors of systems in simulations, and likely in experiments as well. We thus do not agree that our work is ~~simply~~ a mere numerical “extension” of Ref. [14] – it would be so if Ref. [14] is a non-mean-field theoretical calculation in three dimensions. Considering the novelty of the theoretical proposal, and the significance of the numerical examination, we believe that our manuscript is suitable for publication in Nature Communications.

The novelty of the present study, in particular its relation to Refs. [10, 14, 34, 35], has been clarified in the introduction. Please see also our answer to a related comment by Reviewer #3, starting from *On a general ground*, ...

The results provided by the authors are certainly convincing, also considering that they are in line with previous results and theoretical speculations. The numerical approach and the statistical analysis are well detailed in the SI information.

We thank the reviewer for the positive comments.

As a final remark, I notice that the exposition is somehow involved, and that the manuscript is not able to smoothly convey its results. This is particularly true for the section describing Fig. 2. Regardless of the editorial decision on this manuscript, I suggest the authors to review and simply the exposition to better convey their results.

We thank the reviewer for pointing out this weakness. In order to better convey the results in Fig. 2, we have added a schematic plot predicted by the theory (Supplementary Fig. S9), together with a paragraph of explanation (see the paragraph starting with *Connection to the free energy landscape* in Supplementary Note 3). We have also clarified the connection between our results and those in spin glasses (see section *Aging and slow dynamics* in the main text). We believe that these revisions would help a lot the readers to have a better understanding of the data presented in Fig. 2.

From the technical side:

- I do have difficulties understanding why the authors decided to focus on a system of $N = 1000$ particles with diameters distributed according to a power law. Of course, $N = 1000$ is too small for any definition of power law to make sense. I understand that the authors are following previous works, but this is not a good reason for this choice. Perhaps the authors are aware of this point, as at the end of the SI they introduced a section discussing results on a bidisperse two-dimensional system of hard disks.

The purpose of this choice of particle size distribution (Eq. (S2)), rather than common forms such as Gaussian or flat distribution, is to optimize the performance of swap algorithm used to prepare initial configurations, while suppressing crystallization [36]. In this study, we have examined the finite-size effect by changing the system size from $N = 500$ to $N = 8000$. While the same order of N is commonly used in shear simulations of amorphous solids (see for examples, Refs. [2, 3, 7, 11, 13, 15, 16, 18, 19]), we agree that these systems might be too small to accurately define a power law distribution. Unfortunately, accessing to even larger systems is beyond our reachable computational resources. On the other hand, we expect this insufficiency to be already minimized in our results by averaging over many samples whose particle diameters are generated independently according to Eq. (S2). In the modified manuscript, we have clarified our reason for this choice of distribution (see Supplementary Note 1).

- *What is the reason why different configurations at volume fraction φ are obtained compressing the same equilibrium configuration, just changing the initial velocities? Since the equilibrium state is well below the MC density, the system is not going to relax and explore its phase space during compression. Thus, the different configurations at volume fraction φ the paper average over are expected to be extremely correlated.*

The different configurations at φ (called *realizations* in our manuscript), compressed from the same equilibrium configuration (called *sample*) with different initial velocities, are only strongly correlated in the normal glass phase. This can be seen in Supplementary Fig. S6a since the stress-strain curves of different realizations (from the same sample) collapse, for the case $\varphi < \varphi_G$. However, the situation is very different in the Gardner phase: the stress-strain curves are clearly realization-dependent, as shown in Supplementary Fig. S6b. This is the reason why averaging over realizations is needed. Physically, it means that different realizations could end up in different sub-basins in the Gardner phase. We have added the sentence “*note that different realizations may end up in different sub-basins*” in the section *Interpretation of results* in order to clarify this point.

- *What is the criterion used to define the yielding transition? The circles in Fig.1 b and c appear to be placed at arbitrary point. For example, in pane b one could speculate the yielding transition to occur at $\gamma \sim 0.11$.*

The same question was also raised by Reviewer #2 (see point 1). Please see above for a detailed discussion.

Answer to the referee comments

Dear Editors:

We wish to warmly thank you and the reviewers for reviewing again our manuscript. We have addressed the remaining minor suggestions of reviewer #4. The details are provided below and highlighted in red in the accompanying revised manuscript. We hope that the manuscript will be considered appropriate for publication now.

Sincerely,
the authors.

Report of Reviewer #4

The revised version of manuscript NCOMMS-16-25423A better clarified that the manuscript aims at revealing signatures of Gardner transition, which has been predicted to occur at the mean field level, in finite dimensional systems. In particular the manuscript main claim is that the devised numerical protocol used to detect these signatures could be experimentally used.

I am not convinced by this claim, considering that the devised protocol involves the compression of hard-sphere systems, the authors thermalize above the mode-coupling transition using a particle swap algorithm. In order to use this algorithm, the authors are also forced to consider a peculiar system, for which the particle size has a power-law distribution. Experimentally, it is just impossible to thermalize hard-sphere colloidal systems above their mode coupling transition. Therefore, the experimental relevance of these results remains unclear. I'd like also to note that, while the authors stress that their protocol is suitable "to investigate a large variety of experimental systems", they actually only investigate hard-sphere systems. In particular, they investigate hard sphere system close to their jamming transition. Since close the jamming transition these systems are known to have very peculiar properties, e.g. a diverging correlation lengths, an enormous non-affine response, etc., it is quite possible that the observed phenomenology is peculiar to systems close to jamming. Thus, the relevance of these results to other systems is not obvious. Apart from that, as previously mentioned and now better clarified by the authors, the reported results are not particularly novel, as the role of the Gardner transition in the rheological properties of hard-sphere systems has been previously discussed. Thus, while I acknowledge the great numerical effort, it seems to me that the manuscript is neither bringing a novel conceptual understanding of the rheological properties of these systems, nor introducing a truly general and simple protocol to investigate these properties. The relevance of this paper outside the community investigating the Gardner transition in hard-sphere systems is therefore unclear.

The authors have clarified my previous technical concerns, and I do only have few minor suggestions:

- The authors should better explain in the manuscript what it means to shear "quasi-statically" with a constant shear rate > 0 . By definition, quasistatically implies zero shear rate. It is a matter of comparing timescales, and the authors should be clear on this point.

We thank the reviewer for raising the point. To simulate quasi-static shear, we use a very small shear rate $\dot{\gamma} = 10^{-4}$. In the regime of such small $\dot{\gamma}$, the shear rate dependence is negligible in the normal glass phase $\varphi < \varphi_G$, which is demonstrated in Supplementary Fig. S6a. We have clarified this point in the main text.

- Fig. 1 highlights the difference in the rheological properties of systems prepared outside and within the Gardner phase. Unfortunately, the figure compares states that have both a different density and a different pressure. Thus, it is not clear from the figure whereas the observed differences are due to the Gardner transition, or just to the density/pressure. This point is somehow clarified in the Supp. Material, where the authors show stress-strain curves for different values of the control parameters. I would suggest reviewing this figure to compare states having same pressure or volume fraction.

We believe that the reviewer misunderstand our results in Fig. 1. In hard sphere glasses, the pressure is given by $p = p_{\text{glass}}(\varphi, \varphi_g)$, which is the glass equation of state (black dotted lines in Fig. 1), parameterized by the initial density φ_g and the target density φ . As long as we work on the same glass, that is for a given φ_g , it is impossible to have states both outside and within the Gardner phase at the same volume fraction (or pressure), i.e., the state is either inside or outside the Gardner phase at the given φ .